# SARM: Salah Activities Recognition Model Based on Smartphone

**Nafees Ahmad [1]** , **Lansheng Han [1],\*** , **Khalid Iqbal [2]** , **Rashid Ahmad [2]** ,
**Muhammad Adil Abid [3] and Naeem Iqbal [2]**

[1]   School of Computer Science and Technology, Huazhong University of Science and Technology,
    Wuhan 430074, China
[2]   Department of Computer Science, COMSATS University Islamabad, Attock Campus, Attock 43600, Pakistan
[3]   Department of Computer Science and Technology, Shandong University, Jimo, Qingdao 266237, China
*   Correspondence: hanlansheng@hust.edu.cn

**Abstract:** Alzheimer's is a chronic neurodegenerative disease that frequently occurs in many people today. It has a major effect on the routine activities of affected people. Previous advancement in smartphone sensors technology enables us to help people suffering from Alzheimer's. For people in the Muslim community, where it is mandatory to offer prayers five times a day, it may mean that they are struggling in their daily life prayers due to Alzheimer's or lack of concentration. To deal with such a problem, automated mobile sensor-based activity recognition applications can be supportive to design accurate and precise solutions with an objective to direct the Namazi (worshipper). In this paper, a Salah activities recognition model (SARM) using a mobile sensor is proposed with the aim to recognize specific activities, such as Al-Qayam (standing), Ruku (standing to bowing), and Sujud (standing to prostration). This model entails the collection of data, selection and placement of sensor, data preprocessing, segmentation, feature extraction, and classification. The proposed model will provide a stepping edge to develop an application for observing prayer. For these activities' recognition, data sets were collected from ten subjects, and six different features sets were used to get improved results. Extensive experiments were performed to test and validate the model features to train random forest (RF), K-nearest neighbor (KNN), naive Bayes (NB), and decision tree (DT). The predicted average accuracy of RF, KNN, NB, and DT was 97%, 94%, 71.6%, and 95% respectively.

**Keywords:** Salah activities recognition; posture recognition; accelerometer sensor; human activity recognition; classification

## 1. Introduction

Alzheimer's disease (AD) is a disease associated with dementia, in which the patient suffers from prolonged memory loss due to the death of brain cells. Today, an increasing proportion of the population is suffering from AD. There are three stages of AD: early, middle, and late. In the early stage of AD, patients are normally affected through forgetfulness, and losing or misplacing things. For example, a patient may go downstairs for something and then forget why they went there. Treatment costs can be huge to help these patients to carry out their daily activities. According to a survey in the United States, the ratio of people with Alzheimer's varies by age group. The disease starts between the ages of 65 and 69 and affects 5% of the population. This ratio continues to increase with age, reaching 37% for men and 35% for women aged 85 or over [1]. These patients require extra care and help from relatives and caregivers to perform their routine activities. Their disability tends to put extra costs on their care to monitor their condition. Moreover, given today's hustle and bustle and

having so many tasks to juggle in our everyday lives, it is very easy to forget tasks and things in a very normal manner. Even healthy people can experience memory loss in normal routine life.

This is an era of cost-effective technological advancements. For instance, today, mobile phones are commonly used for the recognition of human activities because they usually have incorporated embedded sensors [2]. In particular, among all those sensors, the accelerometer is considered the most feasible to recognize certain human activities [3]. Consequently, there has been a tremendous increase in the use of smartphone sensors to develop healthcare applications, mainly because they are more convenient and economical to be used for research purposes [4]. Many sensor-based applications have been used as medical aids by practitioners and have been used to diagnose patients for various medical reasons, including their heartbeat, motion capture, and blood pressure. These apps have also been used economically to help to monitor the activities of AD patients [5]. They can store patterns of movements and keep track of activities performed and missed, such as eating, opening cupboards, going upstairs, and wandering aimlessly. Sensor-based healthcare applications can help mild AD patients by prompting reminders as well as serving as an alarm for onlookers and relatives of extreme AD patients in case of a sensitive situation, such as falling down the stairs, going to a prohibited area, or skipping meals and/or medicine for a set time [6].

Rapid advances in sensor technology have helped physicians to raise the standard of healthcare services to a great extent. They bring more cost-effective, robust, and accurate activity recognition systems. The emergence of intelligent equipment will help in providing quality surveillance and reminding facilities. Therefore, advanced machines with sensor support can provide extensive healthcare and treatment services to improve life quality. Low-cost multisensors-based smartphones are easily available in the market, through which we can easily recognize the activities. Moreover, smartphone devices have become an important part of everyone's life [7].

Offering prayers (Salah) is a mandatory task for Muslims around the world, and it is compulsory to perform five times a day (Holy Book Quran): *Fajar* (dawn, before sunrise), *Duhur* (afternoon prayer), *Asr* (late afternoon prayer), *Maghrib* (evening prayer (right after sunset)), and *Isha* (night prayer). Around 1.6 billion Muslims throughout the world perform these prayers five times a day. A complete prayer consists of many *Rakat* with prescribed movements, and each *Rakat* refers to a single unit of Islamic prayers with the involvement of such activities (standing (ST), standing to bowing (STB), and standing to prostration (STP)). In each prayer, every person must fulfill a mandatory count of *Rakats*. For example in the Fajar prayer, four *Rakats* need to be performed: first two Sunnah *Rakats* and then two Farad *Rakats*. If the person forgets to perform a *Rakat* between the two *Rakats*, the Salah will not be complete. However, some people may forget the *Rakats* in their prayer. Why is this forgetfulness happening? In today's busy work environment, it is very hard to keep attention and focus on something. Thus, forgetfulness is very common. Similarly, it is difficult to keep track of the *Rakat* parts, in terms of which one was missed and done. This forgetfulness is also associated with early-stage Alzheimer's disease. Hence, these patients need special care to help them to complete their prayer. From this perspective, an Alzheimer's patient or a person with a lack of concentration in prayer can get support with a Salah activities recognition model (SARM)-app based intervention for correction while offering their prayers without any confusion. Thus, the current study is focused on developing a model which would be used to develop an application for the assistance of AD patients to perform the prayer.

Three main steps collectively allow one to complete *Rakat*: (i) standing, (ii) bowing, (iii) prostration (Sujud), which in Arabic are called Al-Qayam, Ruku, and Sujud, respectively (as can be seen in Figure 1). In the present study, we mainly focused on attempts to recognize three set of activities: standing (ST), standing to bowing (STB), and standing to prostration (STP). Correctly performing these three activities can give an indication of correctly completing the *Rakats*. Thus, after the completion of the *Rakats*, the user is able to count their activities to ensure the *Rakats* are completed. If someone is in doubt after performing prayer activities, they will be able to perform them again. In the first stage, we recognized the activities offline. Here, we collected the data with an Android mobile phone and

then performed further operations on the computer to monitor performance using an optimal window length and features of minimum complexity and performed comparisons of three machine learning algorithms in terms of their performance.

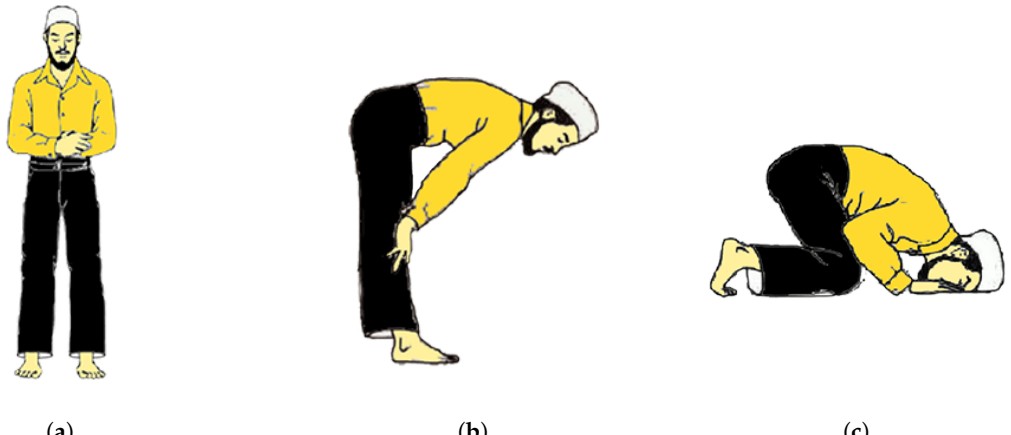

(**a**)  (**b**)  (**c**)

**Figure 1.** Three main activities in each *Rakat*: (**a**) Al-Qayam (Standing); (**b**) Ruku (Standing to Bowing); (**c**) Sujud (Standing to Prostration).

Our presented model's main contributions are:

- To recognize count the three important activities in a *Rakat* of Qayam (ST), Ruku (STB), and Sujood (STP) in prayer of an Alzheimer's patient using six features with low complexity on different window sizes;
- A model that would be used to implement the mobile-based application, which directs the Namazi (worshipper) to perform their prayer according to the code of Islam without any human intervention needed;
- With correct counting and recognition of activities, an Alzheimer's patient can be corrected during prayer with the belief of no mistake or confusion in the prayer on completion.

## 2. Related Work

The human activity recognition concept was first introduced in the 1990s. In the first phase, video-based recognition was studied [8,9]. Many applications for activity recognition have been introduced over the years, such as for healthcare, care of the elderly, to monitor daily living activities, and in security systems [10,11]. In vision-based activity recognition, users' data are gathered from a video and operations are performed to make decisions on video collection frames. Issues with users' privacy, heavy resource utilization, and limitations of monitoring have shifted the attention of researchers away from vision-based recognition to sensor-based recognition. There are two kinds of sensors used: external sensors and mobile-based sensors. A smartphone has a number of built-in sensors, such as an accelerometer, gyroscope, magnetometer, and GPS. Having a low manufacturing cost, along with multiple features, powerful communication capabilities, and fast processing speed make a smartphone very beneficial and useful for activity recognition.

Previously, a lot of work has been done in human activity recognition using embedded sensor technology [1,12,13]. However, limitations exist as researchers focused less on complex activities as compared to simple activities. Therefore, it is still required to explore more human activities with fewer resources utilization to make people's daily lives easier. To support and interpret our idea in detail, we studied the previous work in human activity recognition based on Android mobile phones. Details of the sensors and classifiers used in the previous works are elaborated on in Table 1. Many researchers worked on exercise activities recognition, i.e., walking, running, walking downstairs, etc., [14] and used accelerometer sensors [15,16]. In [16], the author studied six different human activities, namely, slow walking, fast walking, dancing, running, going up stairs, and going down stairs, using six different classifiers and reported achieving an accuracy rate of 91.5%.

**Table 1.** Studies on recognized activities using an Android mobile phone.

| No | Studied Activities | Sensor Type | Machine Learning Classifier | Source |
|---|---|---|---|---|
| 1 | $Ac_1-Ac_{13}$ | Acc | SVM, ANN | [17] |
| 2 | $Ac_0-Ac_2$, $Ac_4$ | Acc | NB, DT, Decision Table | [15] |
| 3 | $Ac_0-Ac_3$, $Ac_4-Ac_6$, $Ac_{13}$, $Ac_{15}-Ac_{20}$ | Acc, LA, Gyro | NB, KNN, DT | [18] |
| 4 | $Ac_0$, $Ac_5$, $Ac_6$, $Ac_{21}$, $Ac_{22}$ | Acc | MP, SVM, RF, SL, LMT | [16] |
| 5 | $Ac_1-Ac_6$ | Acc | SVM | [14] |
| 6 | $Ac_0-Ac_2$, $Ac_4$, $Ac_{14}$, $Ac_{24}$ | Acc | KNN, QDA | [19] |
| 7 | $Ac_1$, $Ac_4-Ac_6$, $Ac_{13}$ | Acc, Gyro, light | DT | [20] |
| 8 | $Ac_0$, $Ac_4$, $Ac_5$ | Acc | DT (C4.5) | [21] |
| 9 | $Ac_0-Ac_2$, $Ac_4$ | Acc | NB, KNN | [22] |
| 10 | $Ac_0$, $Ac_4$, $Ac_{23}$, $Ac_{24}$ | Acc | NB | [23] |
| 11 | $Ac_0$, $Ac_4$, $Ac_{25}-Ac_{27}$ | Acc, GPS, Microphone | Linear Regression, LR | [24] |
| 12 | $Ac_0$, $Ac_4$, $Ac_{13}$, $Ac_{28}$ | Acc | SVM | [25] |
| 13 | $Ac_0$, $Ac_1$, $Ac_4-Ac_6$, $Ac_{29}$ | Acc | ANN | [26] |
| 14 | $Ac_0-Ac_2$, $Ac_4-Ac_6$, $Ac_{14}$, $Ac_{37}$, $Ac_{38}$, $Ac_{29}-Ac_{31}$, $Ac_{24}$ | Acc | SVM, GMM | [27] |
| 15 | $Ac_4-Ac_6$, $Ac_{13}$, $Ac_{14}$ | Acc | SVM, K-mediods, K-means | [28] |
| 16 | $Ac_0-Ac_3$, $Ac_{24}$, $Ac_{32}-Ac_{36}$ | Acc, Gyro | MP, NB, BN, Decision Table, BFT, K-star | [29] |
| 17 | $Ac_4-Ac_6$, $Ac_{13}$, $Ac_{28}$ | Acc, Gyro | KNN, RF, SVM | [30] |
| 18 | $Ac_0$, $Ac_2$, $Ac_4-Ac_6$ | Acc | ANN | [31] |
| 19 | $Ac_0-Ac_4$, $Ac_6$, $Ac_{28}$ | Acc, Gyro, Light, MM, sound level data | NB, KNN, ANN, SVM, DT, RF | [32] |
| 20 | $Ac_0$, $Ac_1$, $Ac_4-Ac_6$, $Ac_{28}$ | Acc | ANN | [33] |
| 21 | $Ac_0-Ac_2$, $Ac_4-Ac_6$ | Acc, GPS, Gyro, MM | NB, SVM, ANN, KNN, LR, RBC, DT | [34] |

**Activities**: $Ac_0$ = Running, $Ac_1$ = Standing, $Ac_2$ = Sitting, $Ac_3$ = Lying, $Ac_4$ = Walking, $Ac_5$ = Walking Upstairs, $Ac_6$ = Walking Downstairs, $Ac_7$ = Standing-to-Sitting, $Ac_8$ = Sitting-to-Standing, $Ac_9$ = Sitting-to-Lying, $Ac_{10}$ = Lying-to-Sitting, $Ac_{11}$ = Standing-to-Lying, $Ac_{12}$ = Lying-to-Standing, $Ac_{13}$ = Jogging, $Ac_{14}$ = Riding a Bike, $Ac_{15}$ = Eating, $Ac_{16}$ = Drinking, $Ac_{17}$ = Smoking, $Ac_{18}$ = Biking, $Ac_{19}$ = Typing, $Ac_{20}$ = Writing, $Ac_{21}$ = Slow Walk, $Ac_{22}$ = Fast Walk, $Ac_{23}$ = Still, $Ac_{24}$ = Driving, $Ac_{25}$ = Stationary, $Ac_{26}$ = Sleep, $Ac_{27}$ = Voice, $Ac_{28}$ = Jumping, $Ac_{29}$ = Hopping, $Ac_{30}$ = Watching TV, $Ac_{31}$ = Vacuuming, $Ac_{32}$ = Climbing, $Ac_{33}$ = Cleaning kitchen, $Ac_{34}$ = Cooking, $Ac_{35}$ = Washing Hands, $Ac_{36}$ = Watering Plants, $Ac_{37}$ = Elevator up, $Ac_{38}$ = Elevator down; **Classifiers**: RF = Random Forests, KNN = K-Nearest Neighbors, LMT = Logistic Model Tree, MP = Multilayer Perceptron, BN = Bayesian network, LR = Logistic Regression, BFT = Best-First Tree, RBC = Rule Based Classifier, DT = Decision Tree, SVM = Support Vector Machine, ANN = Artificial Neural Network, GMM = Gaussian Mixture Model; **Sensors**: Acc = Accelerometer, Gyro = Gyroscope, GPS = Global Position System, MM = Magnetometer, LA = Linear Acceleration.

Two studies were based on transition activities, i.e., going from standing to sitting, sitting to lying [17,35]. In [17], the author recognized 12 different human activities using mobile sensors with the help of Artificial Neural Network ANN and Support Vector Machine SVM. The accuracy of simple activity recognition is high compared to complex human activities. In one study [35], the authors examined many supervised and unsupervised classifiers using three wearable inertial sensors on three body positions to recognize different daily living activities. In this work, 12 different activities with some transition activities were introduced, and the results reported high accuracy for the simple and transition activities. To the best of our knowledge, very few studies have been done on complex activities' recognition. In [18], the author recognized different simple and complex activities using an accelerometer and gyroscope. They reported a 2 s window was enough to recognize simple activities (repetitive activities), like walking and running, but to recognize complex activities (less repetitive), i.e., eating, drinking, and talking, the 2 s window was not enough. Therefore, they used a fusion of sensors data and performed experiments on different window sizes from 2 to 30 s and reported achieving a high accuracy by increasing the window size. However, they performed analysis on a specific and limited age of people, namely, 10 physically fit participants (age range: 23–35 years old), which would make the results biased for unhealthy people and an older age group. According to [27], the feature selection, feature extraction, and the classification algorithm play important roles in achieving the optimum performance. They recognized many simple and complex activities (see Table 1) using kernel discernment analysis with SVM and found an average accuracy of 94%. Similarly, according to the authors of [29], the recognition accuracy of complex activities is poor compared to simple activity recognition. In this work, the maximum accuracy of complex activities was achieved (52% to 72%) by increasing the window size from a single instance to 40 instances. However, they did not perform experiments using the combined data of an accelerometer and gyroscope sensor. In [36], a smartphone-based application was introduced to assist AD patients. In this system, they made applications using the cloud through web services. The authors perceived the environmental data through sensors, then processed the data on the cloud, and an alarm was generated on a mobile in case of an accidental situation. However, the system just monitored the patients with some limitations and could not provide such a facility when the patient was out of range. For example, if a patient went outside for shopping, walking, or to visit some friends, the system was not able to work. To assist and help Alzheimer's patients, Smart Mind [4] was introduced in 2015. The actual motive of this system was to give support to people to perform certain activities and to inform caregivers if they got into a bad situation. This work, however, is limited to specific places, like hospitals or home, and requires heavy resources and high maintenance.

## 3. Salah Activities Recognition Model (SARM)

The proposed system framework is shown in Figure 2. This study entails the collection of data using the sensor, data preprocessing, segmentation, feature extraction, and classification.

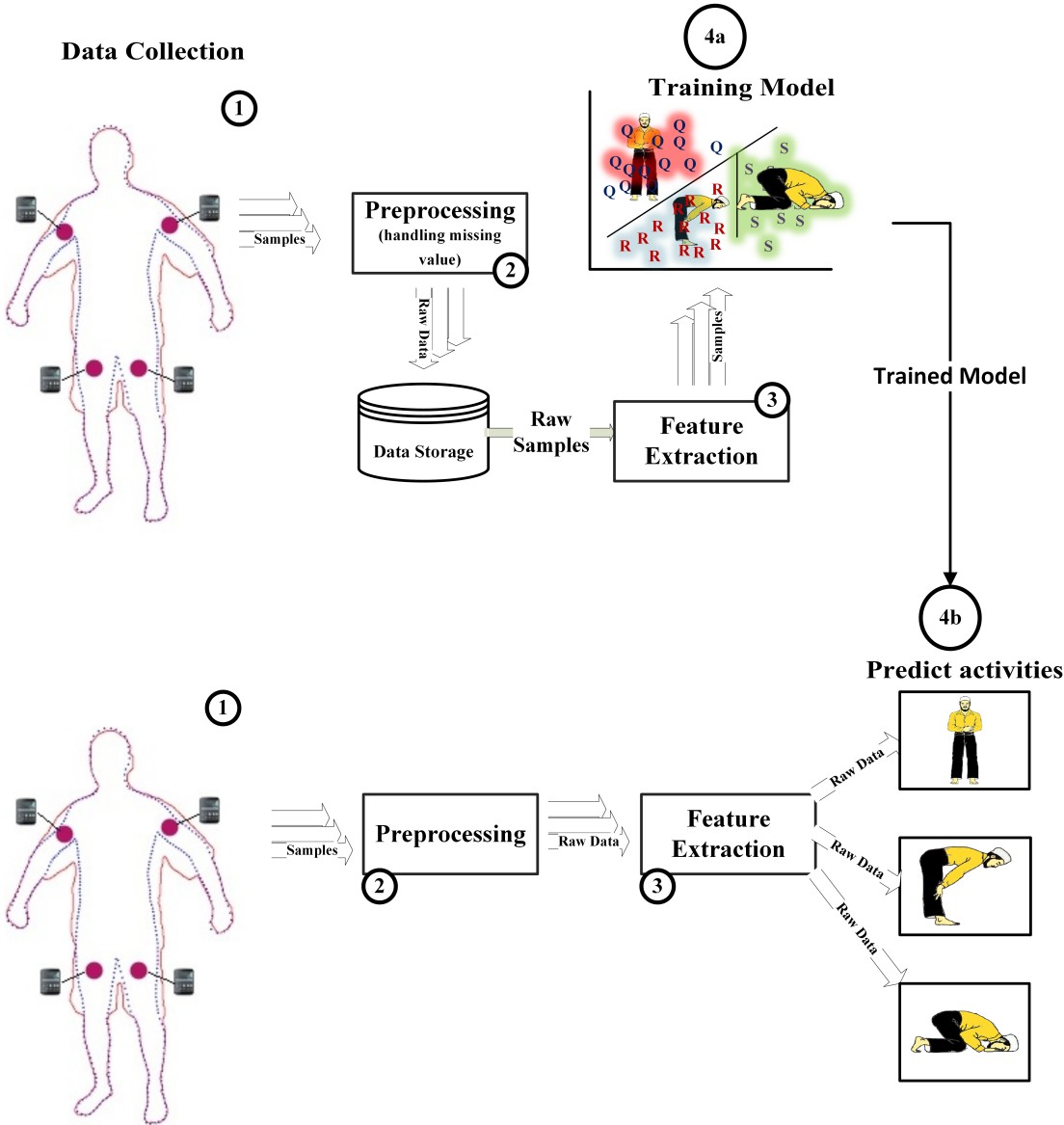

**Figure 2.** Framework of the Salah activities recognition model (SARM).

### 3.1. Data Collection

We proposed the system to recognize the activities in prayer. At the beginning, we had no supply of relevant data for our experimental purpose. Therefore, to achieve our goal, we collected the dataset from multiple candidates. We used the Android MATLAB built-in sensors support package for getting the sensors' values. The laptop MATLAB script was connected with a mobile phone (MATLAB sensor support package) through a local area network. Mainly, five sensors are available in the MATLAB support package, measuring acceleration, orientation, angular velocity, magnetometer, and position. We preferred the accelerometer over the other sensors because it has been widely used in human activity recognition. In practice, our activities were somewhat close to standing and sitting activities, yet the accelerometer sensor recognition accuracy was better to detect these compared to the other sensors [18,37]. Four body positions were considered for data collection: right upper hand, left upper hand, front right trouser pocket, and front left trouser pocket, as shown in Figure 3. Other body positions can also be considered for data recording, like the waist, left lower leg, right lower leg,

and head, but we considered just these positions because people can easily place their smartphone on the upper hands and pockets during the Salah activity. Mobile phones were attached to the abovementioned positions during the collection of data. We recorded three body positions of the test subjects: standing (ST), standing erect to bowing (STB), and standing to prostration (STP) (see Figure 1).

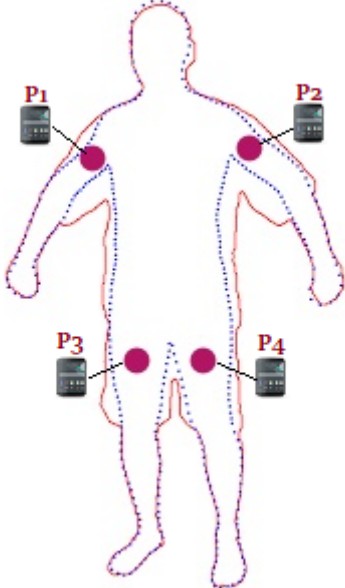

**Figure 3.** Mobile placements (P1, P2, P3, and P4) on the human body during the data collection.

There are certain legitimate ways to offer prayer, which are as follows:

- First, Salah is dependent on recitation. The duration of each activity (ST, STB, and STP) depends on the user (how much time they take to recite) recitation of Salah (prayer). Some recite at a low pace, while others are a bit faster. The estimated duration of ST is from 12 to 20 s. However, for STB and STP, the average spans are from 5 to 9 s.
- The variation is gender-biased; for men, the obligation is to put the hand on the waist and for a female to put the hand on the chest.

In order to meet the requirements, we prepared a diverse dataset. For the collection of data, we considered Muslim students enrolled in the university. Our data set comprised 10 people (age 24–35, weight 55–100 kg, height 150–185 cm). The detail descriptions about the data and subjects are given in Table 2.

**Table 2.** Brief summary of the data collection and details about the subjects. C3, C4, and C5 are columns three, four, and five. S1 to S8 are Males; S9 and S10 are Females

| Subject | Weight (kg) | Height (cm) | Average Time (s) Spent on Single ST ($C_3$) | Average Time (s) Spent on Single STB ($C_4$) | Average Time (s) on Single STP (C5) | Total Time (s) of ST (20 × C3) | Total Time (s) of STB (20 × C4) | Total Time (s) of STP (20 × C5) |
|---|---|---|---|---|---|---|---|---|
| S1 | 90 | 177.8 | 16 | 6 | 6 | 320 | 120 | 120 |
| S2 | 56 | 168 | 15 | 4 | 8 | 300 | 80 | 160 |
| S3 | 65 | 173 | 12 | 5 | 7 | 240 | 100 | 140 |
| S4 | 97 | 183 | 15 | 6 | 8 | 300 | 120 | 160 |
| S5 | 75 | 175 | 18 | 5 | 6 | 360 | 100 | 120 |
| S6 | 85 | 178 | 15 | 7 | 7 | 300 | 140 | 140 |
| S7 | 80 | 178.3 | 17 | 5 | 5 | 340 | 100 | 100 |
| S8 | 90 | 175.2 | 20 | 5 | 8 | 400 | 100 | 160 |
| S9 | 70 | 170.6 | 14 | 5 | 5 | 280 | 100 | 100 |
| S10 | 62 | 172.2 | 16 | 6 | 9 | 320 | 120 | 180 |
| **Total time of recorded data from body position** | | | | | | **3160** | **1080** | **1380** |

The basic procedure which we used to collect the data set was:

- Initially, we connected the mobile phone to the laptop MATLAB cloud.
- We developed a code script for data acquisition:

    1. Initialize mobiledev object.
    2. Enable sensors with sample rate 50 Hz.
    3. Disable sensors.
    4. Save the data into mat files from the log.

- We loaded the code script into the MATLAB cloud.
- All the candidates performed each activity (ST, STB, and STP) 20 times. Every instance of activity was recorded separately. Here, the mobile phone served as a wearable sensor (for data collection) and was deployed on our specimen set of two women and eight men. For all 20 activities, the subjects were briefed first each time they started with the activity, but the order of performing the activity was completely up to the subjects.

    - ST is a standing activity; here, men place their hands on their navel and women place their hands over their chest. As Salah is observed in a whispering mode, we were unable to track the progress and accomplishment of ST. For this reason, we directed our volunteers to rest their hands for a moment before indulging in the second phase. However, these indicative samples were later removed from the experimental data and were not considered to be noted in our findings. The longevity of the interval time for ST depended on the recitation speed of the user (about 12 to 20 s). To record ST, we executed the MATLAB script. When the activity was finished by the performer, we disabled the sensors through the hard code and saved the data. The same procedure was followed to record the ST activity (20 times) of each candidate.
    - In STB, a user moves from a standing position to a bowing position, and there they rest for more or less 4 to 7 s (depending on the user recitation speed), and then move back to the standing position, with this all deemed as one series of STB activity. We executed the script when the user started the activity, and after we were done with recording the first series of STB activity, we disabled the sensor and saved our initial data. In the same way, 20 STB series were recorded for each candidate.
    - In STP, the user moves from a standing position to a prostration position, where they stay for 5 to 9 s, and then after performing the prostration recitation, they continue from prostration to a sitting position, with this total sequence depicting one complete cycle of STP activity. For each user, we recorded the STP activity 20 times.

*3.2. Data Preprocessing*

For data preprocessing, several outcomes were met. Every single instance of activity was dealt with separately. As mentioned earlier (in Section 3.1), we gathered some 200 series patterns from each volunteer. We processed every single pattern manually. At certain times, we noticed that in some cases, the pattern totally deviated from the normal activity pattern. The main reason for this was disconnectivity of the mobile with the laptop. To solve this problem, we removed those patterns from the data. The second challenge was to synchronize the sensor reading (acceleration values (x, y, and z)) with the time range. For data acquisition, our sample rate was 50 Hz, but we received the data with some minor fluctuations. For example, from the sensors, we observed the values for 10 s. In accordance with 50 Hz, the data samples entered should have been 500; however, the values did not come up exactly as 500. Therefore, to make this sensor read data exactly on 50 Hz, we resampled and interpolated the data [38].

*3.3. Segmentation*

The second important step after data preprocessing was segmentation. Segmentation is a crucial process in which the data are divided into chunks for further processing. The accurate chunk size plays a vital role in the detection of activity; it affects the features, and whenever any features get affected,

it directly hinders the performance of a classifier. Three kinds of segmentation could be used in the activity recognition. These are sliding window, activity-defined window, and event-based window segmentation. We used a sliding window technique (windowing) because its implementation is simple and has been widely used in daily activity recognition [39]. In addition, the sliding window technique is beneficial for both static activities and periodic activities (repetitive activities) recognition. For periodic activities, like walking or running, a small size window is enough to get high performance [7,39], but for complex human activities, a small window size cannot accumulate the data samples, and it leads to false results [18,40]. The large window size is expansive in terms of resource utilization, but it gives high performance, so the size of the window should be chosen in this manner, in which each window gives enough readings to enable distinguishing among similar patterns [41]. In our work, the ST activity was quite similar to the standing activity (see Figure 1a), but the usual standing activity lasts for nearly 1 s or more. However, in the ST event, the average person's time is at least 12 s, the maximum being 20 s. On the other hand, STB and STP activities are not regular activities compared to the ST activities discussed earlier. Considering the selection of the optimal window size is problem-dependent and our single activity for ST is from 12 to 20 s (as we discussed in Section 3.1), we extracted features by setting up the window size to different seconds. Initially, we set the window size to 12 s to test high performance, then 14, 16, 18, 20, and 22 s.

Two techniques were employed in the window segmentation: (1) window overlapping with a fixed size window, and (2) non-overlapping with a fixed size window. The window overlapping method greatly leads to high performance but with the disadvantage of high computation needs [33]. Given our motive to achieve high performance, therefore, we selected the window overlapping method, and we overlapped each window with the half samples (50%) of the previous window. The value for overlapping was considered on the basis of previous work [34,42]. The total number of samples overlapped between two consecutive windows can be determined using Equation (1). For example, we fixed our sample rate at 50 and initial window size at 12 seconds; hence, for 50% overlapping in each window out of 600 data samples (shown in Figure 4), exactly 300 values were overlapped from the previous window.

$$NOL = SR \times WL \times \frac{OLP}{100} \tag{1}$$

where $NOL$ is the total number of samples overlapped, $SR$ is the sample rate, $WL$ is the window size, and $OLP$ is the overlapping percentage value.

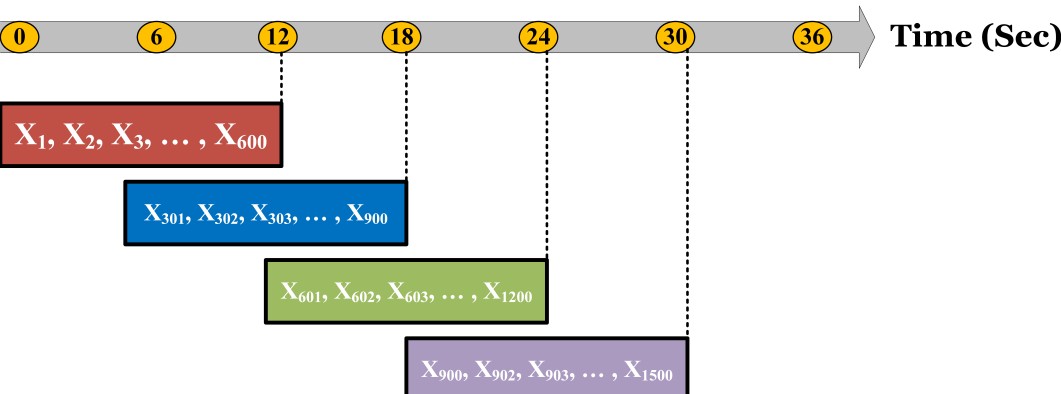

**Figure 4.** A visual representation of a 12 s sliding window data plus 50% overlapping from a previous window.

### 3.4. Feature Extraction

In each slot of sliding window, the data occur in raw form, and passing raw data directly to the classifier may increase the computations. Therefore, after breaking down the data, we drew out different features from each window. Since a specific set of features was not previously identified,

we had to decide which features to use. Ideally, time–domain features will be used because they consumes fewer resources and are widely used for activity recognition [27,35]. Time–domain has many features, but we did not consider all of them. Initially, we manually analyzed the raw data of our activities, especially STB and STP, and computed the maximum for every 12 s segment of data; this greatly improved the performance. Then, we decided to further explore a different set of features that could be used to recognize our activities in a more sophisticated manner. For that reason, we prepared six sets of features (as described in Table 3) on the basis of low computational complexity and storage complexity. The low complexities of these features make them appropriate for mobile phone implementation, as elaborated in Table 4; for more information about the complexity of these features, refer to [43]. The reason for evaluating six sets of features is to achieve different results and to select the best set of features in terms of performance.

**Table 3.** Feature sets and their notations.

| Feature set | Features | Short Notation |
|:---:|:---:|:---:|
| 1 | Max | *fs1* |
| 2 | Max, Min | *fs2* |
| 3 | Max, Min, Median | *fs3* |
| 4 | Max, Min, Median, Mean | *fs4* |
| 5 | Max, Min, Median, Mean, Percentile | *fs5* |
| 6 | Max, Min, Median, Mean, Percentile, Standard Deviation | *fs6* |

**Table 4.** Features and their computational requirements.

| Feature | Computational Cost | Storage Requirement | Suitable for Mobile-Phone |
|:---:|:---:|:---:|:---:|
| Max | very low | very low | Yes |
| Min | very low | very low | Yes |
| Median | medium | very low | Yes |
| Mean | very low | very low | Yes |
| Percentile | very low | very low | Yes |
| Standard deviation | very low | very low | Yes |

Moreover, these sets of features are very simple, and almost all researchers use them for activity recognition [34]. Our set of activities was not so repetitive, and after some moments, a subject's posture changes abruptly, specifically in the STB and STP activity. As max. and min. are the best features to pick up a sudden change from the data, we added them in our feature sets. Here, the point of relevancy is differentiation of the human body postures among STB and STP activities; thus, we chose the median as the best feature to distinguish the postures accurately [44]. Hence, we added median in *fs3*. Moreover, sitting and standing were involved in our activities, and a number of researchers have used the mean feature to identify these postures [45,46]. Therefore, we included the mean feature with the other features in *fs4*. At this stage, we observed a decreasing trend in performance. Thus, we tried percentile in *fs5* to improve performance. In addition, the standard deviation is a more suitable feature to work in combination with the maximum, mean, and minimum features [47]. Thus we added a standard deviation with the *fs5* features to upgrade it to *fs6*.

For a better and simple understanding of our model, we show the algorithm of the proposed system in Algorithm 1 and block diagram below in Figure 5.

---

**Algorithm 1:** SARM

---

$a$ is acceleration samples, $T$ is Time, $RData$ is raw data
$SR$ is sample rate, $RSData$ is re-sample data
$TSW$ is total samples in window, $WL$ is window length
$NOL$ is total number of samples overlapped, $OLP$ is overlap value
$EF$ is extracted feature, $FS$ is Feature set

1　$RData \leftarrow a$ w.r.t $T$　　　　　　　　　　　　　　　　// Collection of dataset
2　$L_i \leftarrow$ Labeling data　　　　　　　　　　　　　// (i = 1, 2, 3), label= ST, STB, STP
3　Remove missing values
4　$T_0 \leftarrow RData.T(start)$
5　$T_{end} \leftarrow RData.T(end)$
6　$T_{new} \leftarrow T_0 : \frac{1}{SR} : T_{end}$
7　$RSData \leftarrow$ Interpolation $(RData.T, RData.a, T_{new})$
8　**for** $s \leftarrow RSData(1)$ to $RSData(end)$ **do**
9　　$TSW \leftarrow SR \times WL$
10　　**if** *first window* **then**
11　　　No overlapping
12　　　**else**
13　　　　$NOL \leftarrow SR \times WL \times \frac{OLP}{100}$
14　　　　Overlap $NOL$ sampled from previous window
15　　　**end**
16　　**end**
17　　**for** $FS \leftarrow 1 \leq 6$ **do**
18　　　$EF \leftarrow F(TSW)$
19　　　$rf \leftarrow RF(EF, L_i)$
20　　　$kn \leftarrow KNN(EF, L_i)$
21　　　$nb \leftarrow NB(EF, L_i)$
22　　　$dt \leftarrow DT(EF, L_i)$
23　　**end**
24　**end**

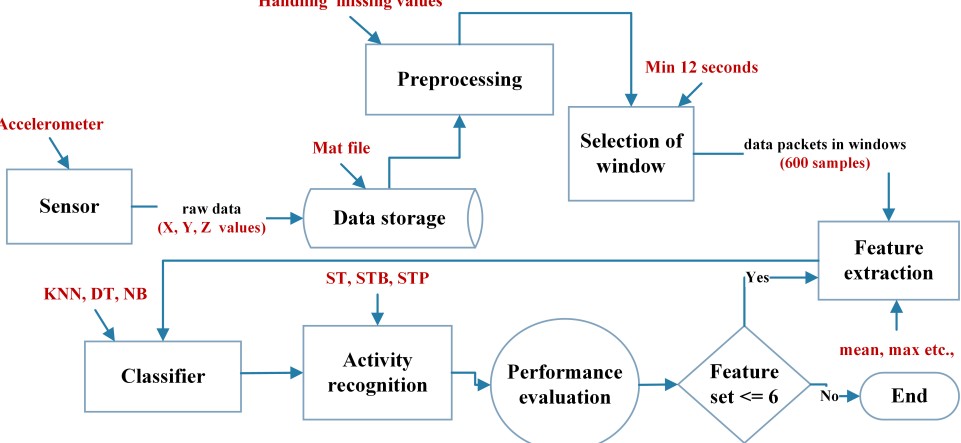

**Figure 5.** Block diagram of SARM.

## 4. Evaluation

We assessed the accuracy of three activities at four human body positions. For each position of the sensor, the results were analyzed in terms of the different feature sets corresponding to multiple window sizes in increment form. For performance evaluation, we implemented our work in MATLAB (2017a) with the help of the Machine Learning toolkit. We chose four classifiers i.e., the random forest (RF), K-nearest neighbor (KNN), decision tree (DT), and naive Bayes (NB) supervised algorithms for classification. These have been widely used in activity recognition and are popular for attaining good results [18,34,39,41]. The RF, DT, NB, and KNN (5 nearest neighbor) classifiers were utilized in the default mode. To train these three classifiers and for performance analysis, we used the K-fold (K = 10) cross-validation technique. In this procedure, data are divided into K equal parts. One subset out of K-subsets is utilized as a test set, while the remaining subsets are used for the purpose of training to create a model. To determine the best performance, this process is run in K iterations. In each run, whole data are reconstructed randomly in K equals parts. In the training phase, data are used as input to the classifier with labels to memorize the activities with corresponding features. While in the testing phase, the unseen data are given to the classifier to predict the best match class. Our observations showed there was no major difference between the performance of the right pocket trouser and left pocket trouser. Therefore, due to similar nature of results, we only considered the right pocket result for discussion. In term of classifiers, RF produced the best results compared to the other three classifiers. Therefore, earlier, we briefly discussed the results of RF classifier with respect to six features sets separately to set the baseline trend. KNN and DT classifiers are observed as the second-best classifier with high accuracy. In the scenario of NB, the results trend were observed to be low. Thereby, we did not include this classifier performance in the main discussion. However, the graphs of NB results are presented in Appendix A.

It ought to be noted that the pockets of the volunteers were not the same during recording of the data, and also the mobile phone was not fixed inside the pockets of the volunteers. That is why the ratio of noise was higher in the trouser pocket data than from the left-hand position and right-hand position data. Thus, we deemed both the left-hand and right-hand positions as more favorable for placement of the sensor. In our work, the recorded dataset was unbalanced because the data of STB and STP were small from the ST activity. This way, the accuracy specifically favors the rich class data over other classes. Thus, we examined the results in the form of precision, recalling, and F-Measure, but mostly, the results analysis is presented in F-Measure because it works with both precision and recall. The following formula was used to calculate the F-Measure:

$$Fm = \frac{((1 + W^2) \times Re \times Pr)}{(W^2 Re + Pr)}, \tag{2}$$

where *Fm* is F-Measure, *Re* is recall, *Pr* is precision, and *W* is a measure that depicts the ratio of importance in both *Re* and *Pr*.

### 4.1. Performance Evaluation of Rf Classifier at Three Body Positions

In this section, we explain the study we conducted to analyze the effect of left-hand, right-hand, and TP body positions on performance. We report the role of these position in the feature sets scenarios on the RF classifier. This classifier was selected for discussion because it has always dominated other classifiers in terms of performance. We used some short notations as right-hand (RH), left-hand (LH), and right pant pocket position (TP). Figure 6 shows the different positions results for each activity in the different feature sets. We have summarized the results of each feature set for all activities considering the subject position. In order to analyze the performance of classifier over the multiple feature sets, we used a 3D graph and confusion matrix to present the results more concisely and clearly.

- The analysis of ST activity in the first feature set resulted in satisfactory performance of 98%–99% on all positions. For the other two activities, the best performance was achieved at the TP position

(96% STB, 98.6% STP). This position leads than the other two positions, with nearly 3% for STB activity and about 2% for STP activity.

- In *fs2*, the average precision decreases relative to FS1. Mainly a drop is observed for STB activity. The corresponding confusion metrics of LH (Table 5), RH (Table 6) and TP (Table 7) indicate that STB activity is mainly confused with ST activity. ST activity has no performance increment or decrement. The performance behavior of ST and STP activities is similar or will vary slightly (See Figure 6c). However, on the *fs2*, the LH position leads (around 92.2%) STB activity over the other two with a margin of 2%.

- By using *fs3*, the overall performance degrades even more, especially for STB and STP activities. This decline can be seen in confusion metrics and in Figure 6c. Mainly, we can see that STB activity is mainly confused with ST activity. However, STP activity is primarily confused with STB activity. In terms of body position, TP is in the leading over STP (93.7%). LH takes the lead over STB (87.7%), and RH leads over ST (98.2%) activity recognition.

- In the case of *fs4*, performance remains the same in most cases. For example, ST activity is recognized high at the RH position, STB at LH, and STP at TP position. However, overall, performance decreased by about 1% to 2% from *fs3*.

- For *fs5*, our experiment has shown improvement due to the addition of percentile feature information with all four features. The trend of the performance went up after continuous decrements, explicitly in favor of STP activity. This improvement can be observed at RH (as shown in the confusion matrix Table 6) and TP position(as shown in the confusion matrix Table 7). Similar, trends are observed for *fs6*.

**Table 5.** The confusion metrics of the random forest (RF) classifier over the 12 s window size on six feature sets at the left-hand (LH) position.

### Predicted Class

| | | Feature Set 1 | | | Feature Set 2 | | | Feature Set 3 | | |
|---|---|---|---|---|---|---|---|---|---|---|
| | | ST | STB | STP | ST | STB | STP | ST | STB | STP |
| | ST | 826 | 8 | 2 | 809 | 5 | 2 | 811 | 0 | 5 |
| Given Class | STB | 3 | 115 | 2 | 12 | 142 | 1 | 14 | 132 | 9 |
| | STP | 7 | 2 | 172 | 9 | 6 | 151 | 12 | 14 | 140 |
| | | Feature Set 4 | | | Feature Set 5 | | | Feature Set 6 | | |
| | ST | 822 | 10 | 9 | 832 | 8 | 2 | 811 | 8 | 0 |
| | STB | 9 | 122 | 10 | 12 | 112 | 10 | 10 | 104 | 7 |
| | STP | 12 | 14 | 129 | 1 | 7 | 153 | 8 | 19 | 170 |

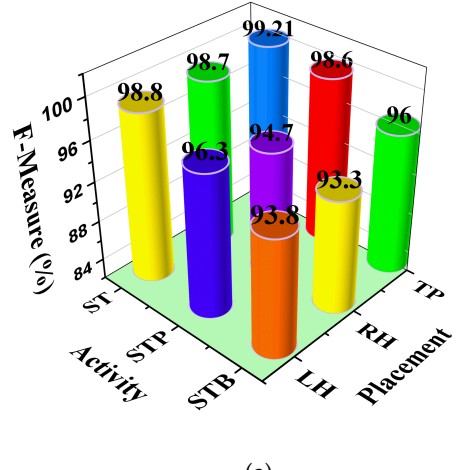

(**a**)

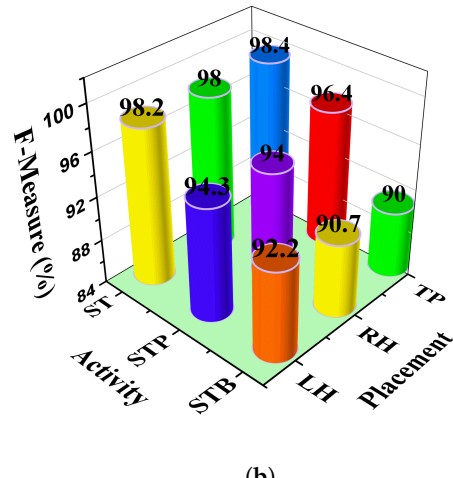

(**b**)

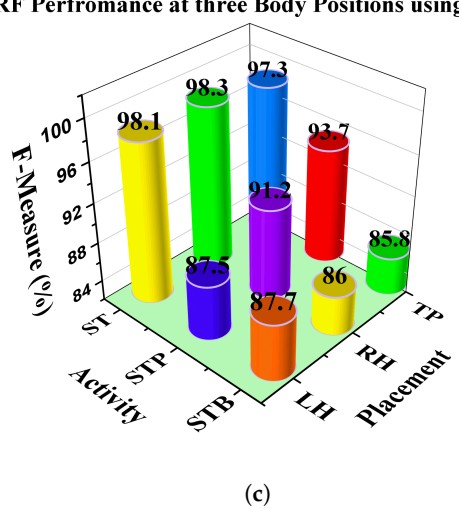

(**c**)

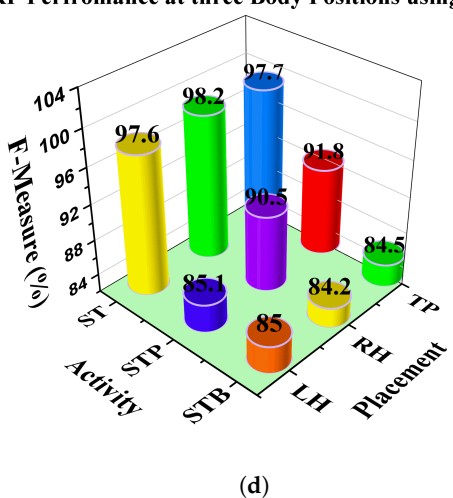

(**d**)

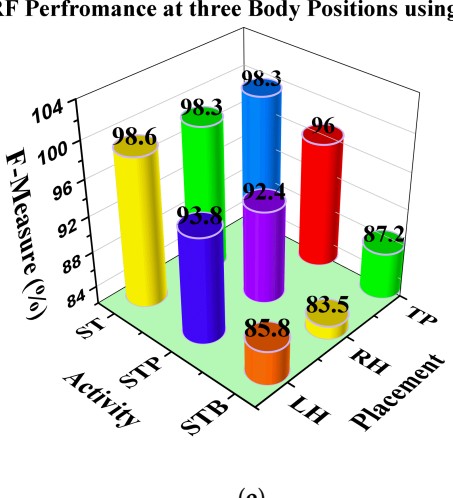

(**e**)

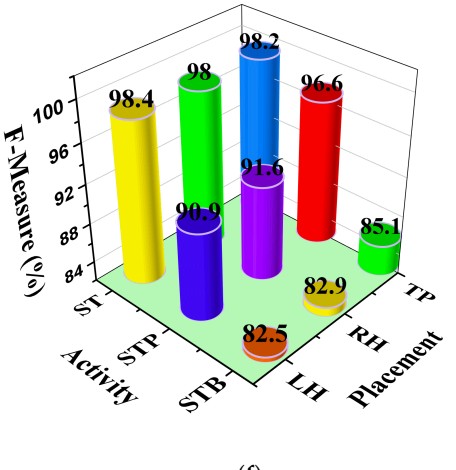

(**f**)

**Figure 6.** Performance of the RF classifier at different positions over the 12 s window size by using: (**a**) feature set 1; (**b**) feature set 2; (**c**) feature set 3; (**d**) feature set 4; (**e**) feature set 5; (**f**) feature set 6.

**Table 6.** The confusion metrics of the random forest classifier over the 12-second window size on six feature sets at the right-hand (RH) position.

**Predicted Class**

| | | Feature Set 1 | | | Feature Set 2 | | | Feature Set 3 | | |
|---|---|---|---|---|---|---|---|---|---|---|
| | | ST | STB | STP | ST | STB | STP | ST | STB | STP |
| | ST | 849 | 5 | 2 | 880 | 6 | 2 | 883 | 5 | 0 |
| Given Class | STB | 7 | 141 | 3 | 15 | 122 | 1 | 16 | 117 | 5 |
| | STP | 8 | 5 | 162 | 12 | 3 | 141 | 9 | 12 | 135 |
| | | Feature Set 4 | | | Feature Set 5 | | | Feature Set 6 | | |
| | ST | 850 | 8 | 4 | 852 | 6 | 0 | 843 | 10 | 4 |
| | STB | 12 | 115 | 14 | 19 | 119 | 10 | 16 | 119 | 14 |
| | STP | 7 | 9 | 163 | 4 | 12 | 160 | 3 | 9 | 164 |

**Table 7.** The confusion metrics of Random Forest classifier over the 12-second window size on six feature sets at the right pant pocket (TP) position.

**Predicted Class**

| | | Feature Set 1 | | | Feature Set 2 | | | Feature Set 3 | | |
|---|---|---|---|---|---|---|---|---|---|---|
| | | ST | STB | STP | ST | STB | STP | ST | STB | STP |
| | ST | 629 | 3 | 0 | 628 | 3 | 1 | 621 | 5 | 0 |
| Given Class | STB | 4 | 98 | 0 | 12 | 91 | 3 | 17 | 85 | 4 |
| | STP | 3 | 1 | 142 | 4 | 2 | 136 | 6 | 2 | 134 |
| | | Feature Set 4 | | | Feature Set 5 | | | Feature Set 6 | | |
| | ST | 643 | 7 | 2 | 649 | 5 | 0 | 630 | 10 | 0 |
| | STB | 13 | 85 | 2 | 13 | 82 | 4 | 12 | 89 | 7 |
| | STP | 7 | 9 | 112 | 4 | 2 | 121 | 0 | 2 | 130 |

### 4.2. Recognition of Activities by Using Knn and DT at Three Body Positions

In Figure 7, we highlighted the results with respect to the different body positions. At each body position, we performed experiments on six features sets, and each feature set was extracted on six different window sizes. Meanwhile, we only discuss the performance of *fs1* and *fs5* on the 12 s window, because compared to other feature sets, *fs1* and *fs5* produced the best results in the form of F-Measure, precision, and recall. Overall, we found that:

- To recognize ST activity using DT on *fs1*, the performance at the RH body position was relatively better than at the other two positions, i.e., 99%. However, the results were satisfactory for the other two positions as well, i.e., 98.3% and 98% at LH and TP, respectively. Using *fs5*, the average performance of overall activities recognition was 1% shifted down with the same classifier.

Figure 7b shows the performance reduction on *fs5* compared to *fs1* at the RH and TP body positions. However, using the feature set *fs5*, we can see that the performance has been improved at the position of LH relative to the other two positions, i.e., 98.28%. Therefore, the performance difference between the RH and LH positions was measured as about 1.08% and at (LH–TP), the difference was about 0.84%.

In the case of the KNN classifier, the performance was not so changed compared to the DT results for ST activity. However, slightly more differences were observed in performance among the body positions on *fs1*. For example, the leading performance was acquired at the RH position, around 98.60%; whereas on the positions of LF and TP, the F-Measure values were about 96% and 97%. In the case of *fs5* using the KNN classifier, the performance was improved at the LH and TP body positions comparatively when we applied *fs1* (on *fs1*, the F-Measure was about 96.87% but on *fs5*, it was about 98%; further, the same trend was observed on TP. For more detail, the performance is shown in the graphs in Figure 7).

- To recognize STB activity in the perspective of the classifiers, the overall performance of DT was better than the KNN classifier. Moreover, in terms of *fs1*, the recognition performance was better at the RH position than the other two body positions using DT. However, using *fs5*, the performance accuracy trend was surprisingly changed at RH towards lower, at about 73.22%, as can be seen in Figure 7b. Therefore, we can say that *fs1* can be a better option to recognize the STB activity instead of *fs5*. However, when we did the comparisons of body positions using *fs5* followed by DT, the performances at TP and LH were observed to be significantly improved compared to the RH position (TP = 82.68%, LH = 83.33%). In the case of the KNN classifier, the STB activity recognition was seen as high at the RH position using *fs1*. while for the performance of KNN on *fs5*, the TP position played the lead role position, i.e., 83.89%, and this time, the result at LH can be seen to be the lowest, about 8.16%, as seen in Figure 7d.

- For STP activity, the trend in performance was changed as compared to the other activities at different body positions, whereby a high recognition performance was achieved for ST and STB at the RH position, but this time, the trend of increment was in the favor of the TP position in all cases. In terms of the role of the classifiers, KNN produced more reasonable results than DT using *fs5*, while DT performed better on *fs1*. The performance differences between DT and KNN on *fs1* were about 0.99% RH, 2% LH, and 1% at the TP positions, respectively.

In conclusion, the ST activity's finest recognition performance was obtained at the RH position in most of the cases (of feature sets and classifiers). Similarly, the RH position was observed to be the most suitable place for STB activity recognition using *fs1*. However, we observed the worst performance for STB activity at the RH position on *fs5*. Moreover, we observed a performance improvement for STP activity at the TP position through KNN classifier on *fs5*. To show the performance of each activity in a simple form, confusion metrics are presented for both classifiers using *fs1* and *fs5* for each body position. Tables 8–10 describe the confusion metrics of DT classifier for the body positions of LH, RH, and TP, respectively, while Tables 11–13 present the confusion metrics of the KNN classifier for the LH, RH, and TP body positions.

**Table 8.** Confusion metrics for *fs1* and *fs5* at the RH position achieved through decision tree (DT).

| | | Predicted Class | | | | | | | |
|---|---|---|---|---|---|---|---|---|---|
| | | Feature Set 1 | | | | Feature Set 5 | | | |
| | | ST | STB | STP | Precision (%) | ST | STB | STP | Precision (%) |
| Given Class | ST | **846** | 7 | 0 | 99.17 | **835** | 18 | 0 | 97.88 |
| | STB | 6 | **133** | 1 | 95 | 23 | **108** | 9 | 77.14 |
| | STP | 6 | 11 | **172** | 91 | 7 | 29 | **153** | 80.95 |
| | Recall(%) | 98.60 | 88.07 | 99.42 | 95.06 | 95.53 | 69.67 | 94.33 | 85.32 |

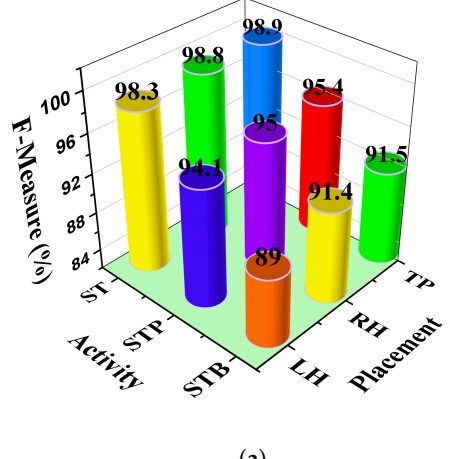

(**a**)

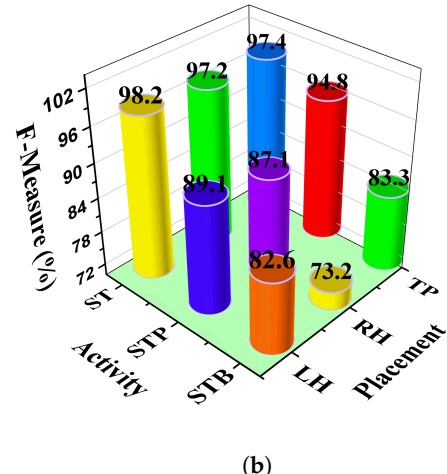

(**b**)

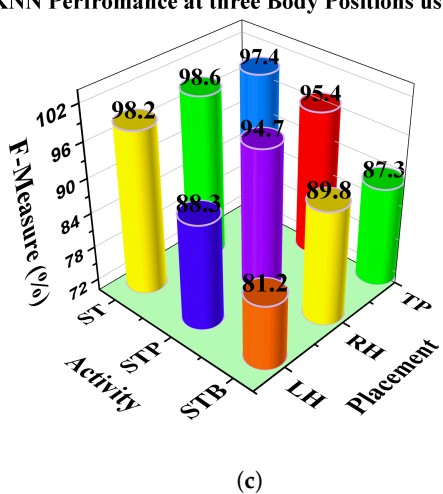

(**c**)

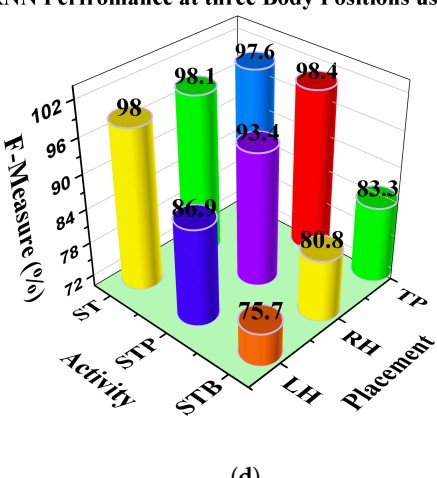

(**d**)

**Figure 7.** Recognition of activities at three body positions using DT and K-nearest neighbor (KNN) at the 12 s window
.

**Table 9.** Confusion metrics for *fs1* and *fs5* at the LH position achieved through DT.

| | | Predicted Class | | | | | | | |
|---|---|---|---|---|---|---|---|---|---|
| | | Feature Set 1 | | | | Feature Set 5 | | | |
| | | ST | STB | STP | Precision (%) | ST | STB | STP | Precision (%) |
| Given Class | ST | **810** | 4 | 1 | 96.38 | **801** | 10 | 7 | 97.92 |
| | STB | 14 | **118** | 5 | 86.13 | 9 | **117** | 20 | 80.13 |
| | STP | 9 | 6 | **170** | 91.89 | 2 | 10 | **161** | 93.06 |
| | Recall (%) | 97.23 | 92.18 | 96.59 | 92.46 | 98.64 | 85.40 | 85.63 | 90.37 |

**Table 10.** Confusion metrics for *fs1* and *fs5* at the TP position achieved through DT.

| | | Predicted Class | | | | | | | |
| --- | --- | --- | --- | --- | --- | --- | --- | --- | --- |
| | | Feature Set 1 | | | | Feature Set 5 | | | |
| | | ST | STB | STP | Precision (%) | ST | STB | STP | Precision (%) |
| Given Class | ST | **626** | 5 | 1 | 99.05 | **630** | 14 | 2 | 97.52 |
| | STB | 4 | **97** | 1 | 95.09 | 14 | **90** | 3 | 84.11 |
| | STP | 3 | 8 | **135** | 92.46 | 3 | 5 | **119** | 93.70 |
| | Recall (%) | 98.89 | 88.18 | 98.54 | 95.53 | 97.37 | 82.56 | 95.96 | 91.77 |

**Table 11.** Confusion metrics for *fs1* and *fs5* at the RH position achieved through KNN.

| | | Predicted Class | | | | | | | |
| --- | --- | --- | --- | --- | --- | --- | --- | --- | --- |
| | | Feature Set 1 | | | | Feature Set 5 | | | |
| | | ST | STB | STP | Precision (%) | ST | STB | STP | Precision (%) |
| Given Class | ST | **847** | 6 | 0 | 99.29 | **842** | 7 | 2 | 98.94 |
| | STB | 11 | **128** | 1 | 91.42 | 22 | **108** | 10 | 77.14 |
| | STP | 7 | 11 | **171** | 90.47 | 1 | 12 | **178** | 93.19 |
| | Recall (%) | 97.91 | 88.27 | 99.41 | 93.73 | 97.34 | 85.03 | 93.68 | 89.75 |

**Table 12.** Confusion metrics for *fs1* and *fs5* at the LH position achieved through KNN.

| | | Predicted Class | | | | | | | |
| --- | --- | --- | --- | --- | --- | --- | --- | --- | --- |
| | | Feature Set 1 | | | | Feature Set 5 | | | |
| | | ST | STB | STP | Precision (%) | ST | STB | STP | Precision (%) |
| Given Class | ST | **805** | 7 | 3 | 98.77 | **809** | 7 | 2 | 98.89 |
| | STB | 23 | **106** | 8 | 77.37 | 19 | **103** | 24 | 70.54 |
| | STP | 19 | 11 | **155** | 83.78 | 4 | 16 | **153** | 88.43 |
| | Recall (%) | 95.04 | 85.48 | 93.37 | 86.64 | 97.23 | 81.74 | 85.47 | 85.96 |

**Table 13.** Confusion metrics for *fs1* and *fs5* at the TP position achieved through KNN.

| | | Predicted Class | | | | | | | |
| --- | --- | --- | --- | --- | --- | --- | --- | --- | --- |
| | | Feature Set 1 | | | | Feature Set 5 | | | |
| | | ST | STB | STP | Precision (%) | ST | STB | STP | Precision (%) |
| Given Class | ST | **623** | 2 | 1 | 99.52 | **637** | 9 | 0 | 98.60 |
| | STB | 24 | **104** | 0 | 81.25 | 21 | **85** | 1 | 79.43 |
| | STP | 6 | 4 | **116** | 92.06 | 0 | 3 | **124** | 97.63 |
| | Recall (%) | 95.40 | 95.54 | 99.14 | 90.94 | 96.80 | 87.62 | 89.2 | 81.89 |

*4.3. Influence of Window Size on the Recognition of Activities*

In this part, we discuss the role of different window sizes on the performance of our set of activities. Since, RF performed well comparatively KNN and DT classifiers. Therefore, window sizes were only examined in the context of KNN and DT. Moreover, the performance of the classifiers were satisfied on *fs1* as compared to *fs5* for all activities. Thus, we performed experiments on *fs5* using different window sizes. Furthermore, in all the feature sets except feature set *fs5*, the result performance was observed in three ways (increase, decrease, same) when we applied different window sizes. However, when we evaluated the role of different window sizes in the scenario of *fs5*, most of the time, the trend of performance was improved. In [29,39], the authors briefly discussed the impact of a large window size and confirmed the overall improvement in the large size window, especially in the case of those complex activities that have shown less accuracy. In our situation, we obtained good results for ST and STP activity, but in the case

of STB, the results were not so impressive. Two reasons may lie behind this low performance: One is the data for STB activity are small compared to those of other activities, and second, this activity has minor complexity. Therefore, we decided to check the effect of a large-size window on performance. The impact of the window size on the three body positions using the KNN and DT classifier are presented in Figure 8. As can be seen, most of the time (14, 16, 18, 20, and 22 s windows), the results were improved at all body positions with both classifiers, specifically for STB activity. The important improvement for STB activity was about more than 30% at the RH and 22% at the LH body positions using KNN, while less improvement was observed at the TP position, at about 7%, because it already had a high accuracy at the minimum window size. A similar improvement can be observed with the DT classifier at all positions. On the other hand, important improvements were observed for STP activity with a high window size at the LH (6%) and RH (10%) body positions using KNN. However, at the TP position, the performance was a little decreased, around about 2% to 3%, under the effect of the large window size. In the case of the DT classifier on STP activity, important improvements were observed for all positions, around 10%, 7%, and 9% at the RH, LH, and TP positions, respectively. The performance at the TP position showed minor fluctuations, indicating sometime it was in increment and sometimes a decrement. According to our observation at that body position, the mobile phone was not fixed at one place, and it was possible for it to shake inside the pocket during the recording of the data, so that could be the cause for the drop in performance.

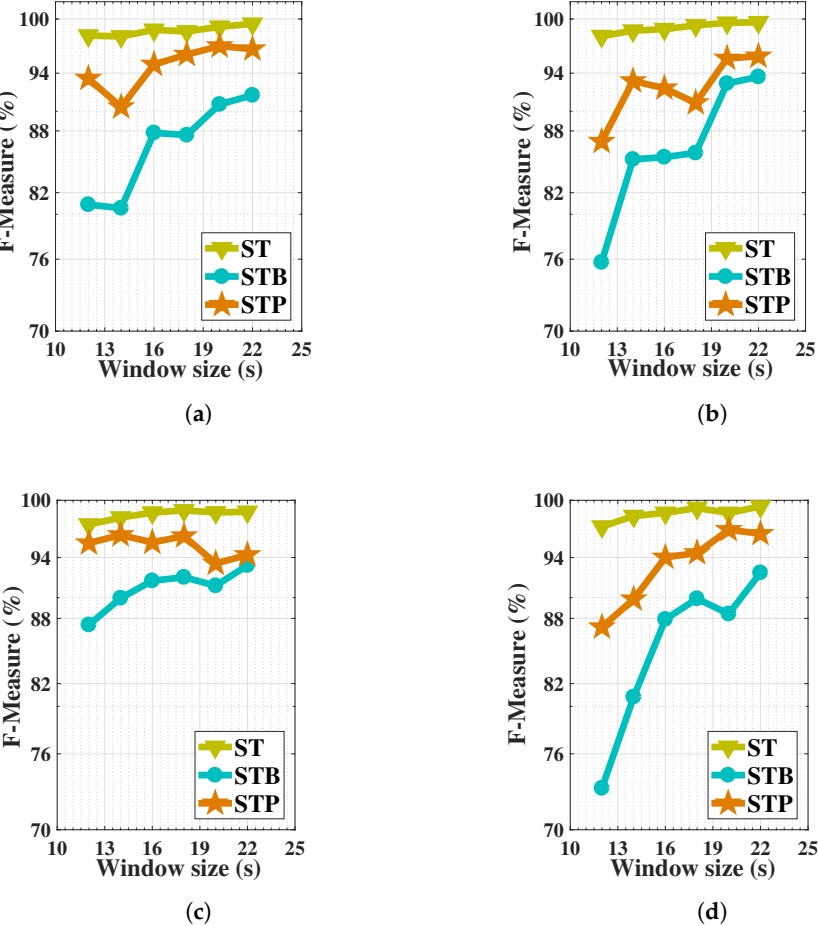

**Figure 8.** *Cont.*

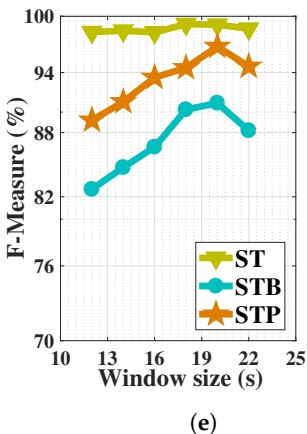
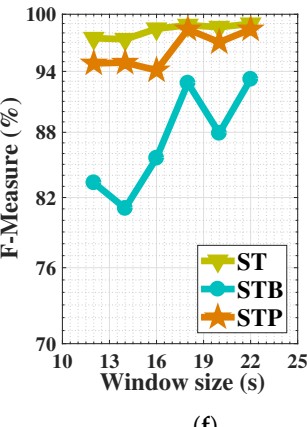

|   |   |
|---|---|
| (**e**) | (**f**) |

**Figure 8.** The impact of the long-size window on activities at: (**a**) the RH position using KNN; (**b**) the LH position using KNN; (**c**) the TP position using KNN; (**d**) RH using DT; (**e**) the LH position using DT; (**f**) TP using DT.

## 5. Conclusions

In this paper, SARM is presented to recognize the activties of prayer (Salah). This study defines the complete approach to implement SARM; in particular, it involved a data collection procedure, preprocessing of data, feature extraction, and training of classifiers. Moreover, six feature sets were applied on the raw data and on each feature set, six windows were applied. Further, we showed the results in terms of evenly incrementing the window size, whereby generally, the trend of performance was seen toward an increment at each body position, while performance loss were observed on some window sizes. Considering the results of classifiers, RF classifier produced the best results in all scenarios than other classifiers. Moreover, the KNN and DT classifiers outperformed on the selected features compared to NB. All the methodology was employed on four body positions, i.e, left-hand, right-hand, trouser right pocket, and trouser left pocket. It is observed that the pockets of the volunteers were not the same during recording of the data, and also the mobile phone was not fixed inside the pockets of the volunteers. That is why the ratio of noise was higher in the trouser pocket data than from the LH and RH data. Thus, we deemed both the LH and RH positions as more favorable for placement of the sensor. Moreover, we observed the different performance ratios for the feature sets, sometime increments and sometime decrements. Probably, the reason is we could not try more and different combinations of features. Therefore, there is a gap in our work, and further optimization in feature selection may further improve the performance.

## Appendix A. Recognition of Activities by Using Naive Bayes at Three Body Positions

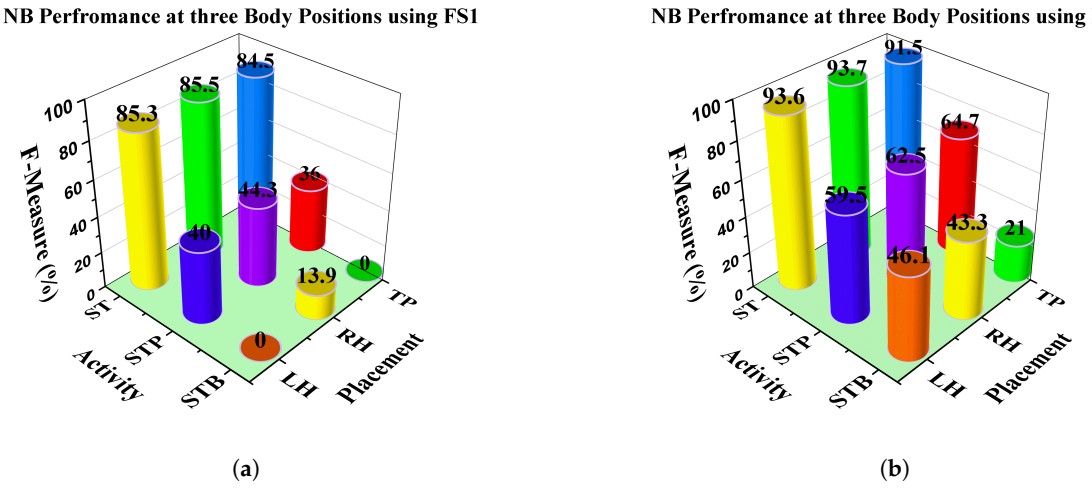

(**a**)　　　　　　　　　　　　　　　　　　　　　　(**b**)

**Figure A1.** Recognition of activities at three body positions using NB.

**Author Contributions:** N.A. produced the main idea, implemented all methodology, and prepared the manuscript. He collected the data from the subjects and evaluated all the data in MATLAB tool. L.H. supervised this work. K.I. and R.A. revised this paper and provided helpful comments and advice. M.A.A. provided his contribution in the shape of a better result presentation. N.I. provided helpful suggestions in the revision of the paper.

**Funding:** This work is supported by the National Natural Science Fund of China NSF: 61272033.y

**Conflicts of Interest:** The authors declare no conflict of interest.

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
