# Peer review of "SARM: Salah Activities Recognition Model Based on Smartphone"

_electronics, doi:10.3390/electronics8080881_

Round 1
Reviewer 1 Report
In this paper the authors study three different data-classification algorithms (K nearest neighbours, Decision trees and Naive Bayes algorithms) in order to classify the user posture from time-domain accelerations measurement, using a smartphone internal accelerometers measurement as input data. The postures recognition is focused on three main posture of "Salah activities" from Muslim prayers rites: Standing, Standing to Bowing and Standing to Prostration. The main contribution of this paper is the application of well-known data-processing and data-classification techniques to yet unexplored activities. The future goal is to implement those algorithms inline on mobile phone embedded application, which would allow real-time feedback to help worshippers during their payer activities.
This paper describes quite precisely the motivations behind its subject and the procedure followed to reach the desired goal. The English style is perfectible and the reader would appreciate a major substantial editing, especially regarding some confusing sentences and non-explicit acronyms (see hereafter for more details). The paper construction respects the required format, with a minor lack of perspectives for further work during the conclusion. The results discussion and conclusion could also benefit from critical point of view regarding the experiments and chosen technologies. On the technical side, the selected solutions consists mainly in the integration of pre-implemented algorithms and communications with MATLAB for data capture and offline processing. Considering this point, the paper could be seen as a preliminary study during the authors learning process with promising perspectives. The authors however could push the reasoning forward to induce a feeling of completeness.
Detailed review:
Keywords list is not consistent with delimiters, starting with ';' and finishing with ','. The keyword "posture recognition" is also used twice.
Overall English style varies from correct to difficult to understand, with some odd constructions e.g. :
- l. 27-28, "This ratio keeps on advancing with age and is at its peak among 85 years old or over and has risen to a fatal 37%",
- l. 29, "carrying out their daily activities" repeats the l. 25,
- l. 34, "Now is also an era…",
- l. 38-39, "cheap, handy and are very economical" seems also odd.
- l. 50-56, the formulations are confusing,
- l. 58-59, "…before the sun rising to the rising sun time."
More on the content:
- l. 49, "error-free" for a computer system doesn't feel like a reachable goal, maybe another adjective should be used.
- l. 58-64: Are "FAJAR", "DUHUR", "ASR", "MAGHRIB", "ISHA" and "RAKAT" acronyms? If so, maybe add a full definition, else, it should not use uppercase (italic is generally preferred to emphasise foreign words), except if it is mandatory in Muslim culture.
- l. 63, "…it is difficult for those who have an illness to the RAKAT…", don't you mean it is easy to forget, or difficult to remember/keep track of?
- l. 71, a typo "would be useD"
- l. 114, "Only two studies were based on transition activities" : Do you mean no other studies exists, or only two from your bibliography?
- Table 1: All acronyms are not explicitly detailed (SVM, ANN, GMM…). Some are doubled (no. 2: DT, Decision table…).
- l. 122-126: Enumerating the whole 13 may not be relevant for the article,
- l. 124-126: From my knowledge, accelerometers measure linear accelerations. Thus both should not be separated, or the difference should be explicitly described,
- l. 148: "…and requires heavy resources for high maintenance.", are the resources required FOR the maintenance, or does it requires resources and maintenance?
- l. 153: "activities in prayer (Salat)", is it a typo for "Salah"?, if it is, maybe the precision is unnecessary as the association salah <-> activities in prayer is already done in previous sections.
- l. 159: You chose accelerometers, which are good for steady orientations measurement, however, maybe your classifier could benefit from the addition of gyroscope data during transition activities,
- Table 2: You specify variables names C3, C4, C5 and S1 to S8 but never use them during the article,
- l. 224: You resampled and interpolated the data. Do you think this process may have side-effects on the processed data? When integrated in real-time the algorithms will access the data without resampling.
- Section 3.3 (no line numbers in the first paragraph): You use the term "sliding window", which commonly implies the data are processed over the window on each new data input, to designate an overlapping window processing. This solution is compute intensive on a 12 seconds window, as you reminds at the end of the paragraph. Maybe you should use a less confusing term to prevent a misunderstanding.
- l. 227-228: "The total number of samples…equation 1" adds no information when compared to the previous sentence.
- l. 231: "almost 300 values" : If your window size is 600 samples and you overlap by 50%, exactly 300 samples overlap the previous window.
- l. 236-238: the reader may be confused by the formulation
- l. 239: "max. feature", the abbreviation style may be inappropriate, and the term is unclear (maximum of what?)
- l. 243: Table 4 is refereed before Table 3, consider adjusting the tables order.
- l. 244-246: the formulation is complicated, it is difficult to understand which subject the verbs relate to at the end.
- Algorithm 1: l.2 L_i <- Labeling data, i=[1-3], what are the labels?
- Figure 5: Between "Classification Algorithm", "Feature Extraction" and "Feature Sets <= 6", lines are overlapping and the directions are confusing.
- l.?? (Section 4. 4th line) : "We choose…", you should use the past time to stay coherent with the rest of the document.
- l.?? : "on the basis of…", might be reformulated
- l.?? : "These are well versed regarding performance in activity recognition.", The formulation feels odd,
- l.?? : You discard the Naive Bayes algorithm without giving any measurable justification
- l. 280: "approx.", abbreviations should be avoided
- l. 293-298: The measurements are given with a 4 digits format, given the relatively small data set, are all the digits significant?
- l. 314: You may add the reference to tables numbers
- l. 329: There seems to be an undefined reference for Figure ??
- l. 330: "both classifiers specifically for SBT activity". maybe a comma is missing
- l. 331: "approx. more than 22%, 30%", is odd, you might prefer a range, or a threshold value.
- l. 331-333: The sentence is too long and hard to understand,
- l. 336: "approx. 6% and 10%". same as for l. 331, you give two values, maybe a range? or specify what variables each value relates to
- l. 350: "some minor decrements", maybe "decreasing value", "decreasing slope", "performances loss"…
- l. 352: The NB is depicted as outperformed by other solutions, but results are never compared or given, which is quite intriguing
- l. 352: "ledft", a typo -> "left"
- l. 357-358: You justify the observed fluctuations in the performances versus feature sets with the fact you couldn't try more features sets, could you develop? What was limiting you?
- l. 360: Could you detail the perspectives, which clues are the more interesting to explore in your opinion if you want to optimize detection performances?
As general questions:
- To generate your datasets, you used LAN connected smartphone streaming sensors data to your computer, leading to the need of pre-processing the data. Could MATLAB app on android save the data locally? This would give better quality datasets.
- Have you considered using multiple time scale features, using convolutionnal neural networks as an inspiration, to address the duration variations of your activities?
- Do you think multiple classifiers in parallel, with a voting system of some sort (like decision forests), could improve the detection performances while still being computationally efficient?
- Do you have an idea of the computational power needed to implement these algorithms in real-time inside the phone?
- How does your algorithm reacts when presented with other activities than the ones it classifies? Is there a potential for mis-detection during the prayer, disqualifying the algorithm for real-life usage?
Author Response
Dear Editor and Reviewer,
Please find the attached Reply letter in response of first reviewer comments. I would like to special thanks to Editor and Reviewer for their thoughtful comments and suggestion.
Sincerely yours.
Nafees Ahmad

Reviewer 2 Report
In this paper, authors proposed salah activities recognition model. Although the proposed work seems to be novel, authors do not present any comparision results to the related work.
In order to demonstrate that the proposed work outperforms the related work, the comparison results should be included in the paper.
Author Response
Dear Editor and Reviewer,
Please find the attached Reply letter in response of second reviewer comments. I would like to special thanks to Editor and Reviewer for their thoughtful comments and suggestion.
Sincerely yours.
Nafees Ahmad

Reviewer 3 Report
1) The tab position is needed not only at the beginning of the section, but at the beginning of each paragraph (lines 34, 48, 57 and so forth).2) Line 85 contains probably a misprint: “sujood”?
3) Lines 115, 151 contains unnecessary tab position at the beginning.
4) Line 153 contains probably a misprint: “Salat”?
5) Lines 185, 221, 223, 224 contains a error: instead “HZ” should be “Hz”.
6) The caption to Figure 4 contains a misprint.
7) The captions to Figure 6 contains probably a misprint: “FS1”, “FS5”?
8) Tables 5-10 contains a misprint in left part: instead “Precision” should be “Precision(%)”.
9) Line 328 contains a misprint: “Figure ??”.
10) Line 352 contains a misprint: “ledft”.
11) Line 357 contains a misprint: “Moreover,We have”.
Author Response
Dear Editor and Reviewer
Please find the attached Reply letter in response of third reviewer comments. I would like to special thanks to Editor and Reviewer for their thoughtful comments and suggestion.
Sincerely yours.
Nafees Ahmad
